# Methyl Donors Reduce Cell Proliferation by Diminishing Erk-Signaling and NFkB Levels, While Increasing E-Cadherin Expression in Panc-1 Cell Line

**DOI:** 10.3390/ijms23052546

**Published:** 2022-02-25

**Authors:** Eva Kiss, Gertrud Forika, Magdolna Dank, Tibor Krenacs, Zsuzsanna Nemeth

**Affiliations:** 11st Department of Internal Medicine and Oncology, Oncology Profile, Semmelweis University, 1083 Budapest, Hungary; kiss.eva3@med.semmelweis-univ.hu (E.K.); titkarsag.dank@med.semmelweis-univ.hu (M.D.); 21st Department of Pathology and Experimental Cancer Research, Semmelweis University, 1085 Budapest, Hungary; forika.gertrud@med.semmelweis-univ.hu (G.F.); krenacs.tibor@med.semmelweis-univ.hu (T.K.); 31st Department of Internal Medicine and Oncology, Semmelweis University, 1083 Budapest, Hungary

**Keywords:** methyl donors, cell cycle, apoptosis, E-cadherin

## Abstract

Pancreatic cancer is an aggressive malignancy with high metastatic potential. There are several lifestyle-related determinants in its etiology, including diet. Methyl donors are dietary micronutrients which play an important role in fueling vital metabolic pathways, and as bioactive food components provide methyl groups as substrates and cofactors. The imbalanced nutritional status of methyl donors has recently been linked to pathological conditions. Therefore, we hypothesized that dietary methyl donors may improve the physiology of cancer patients, including those with pancreatic cancer, and could be used for intervention therapy. In this study, methyl-donor treatment (L-methionine, choline chloride, folic acid and vitamin B12) of an aggressive pancreatic adenocarcinoma cell line (Panc-1) resulted in significantly increased p21^WAF1/Cip1^ cyclin-dependent kinase inhibitor levels, along with apoptotic SubG1 fractions. At the same time, phospho-Erk1/2 levels and proliferation rate were significantly reduced. Though methyl-donor treatments also increased the pro-apoptotic protein Bak, Puma and Caspase-9, it failed to elevate cleaved Caspase-3 levels. In addition, the treatment significantly reduced the production of the pro-inflammatory cytokine IL-17a and the transcription factor NFkB. Similarly, a significant decrease in VEGF and SDF-1a levels were detected, which may indicate reduced metastatic potential. As expected, E-cadherin expression was inversely associated with these changes, showing elevated expression after methyl-donor treatment. In summary, we found that methyl donors may have the potential to reduce aggressive and proliferative phenotype of Panc-1 cells. This suggests a promising role of dietary methyl donors for complementing relevant cancer therapies, even in treatment-resistant pancreatic adenocarcinomas.

## 1. Introduction

The incidence and mortality rate of pancreatic cancer is growing worldwide, and it is expected that it will be ranked the third leading cause of cancer death by 2025 [1]. Pancreatic cancer is an aggressive disease with poor prognosis because of the early presence of metastases, treatment resistance, multiple genetic and epigenetic changes, as well as the fact that it is usually diagnosed at advanced stages [2,3]. All of these contribute to the only 10% 5-year survival rate. In addition to non-modifiable risk factors, lifestyle habits such as smoking, alcohol usage, diet, obesity, and physical activity have been identified as determinant in the etiology of pancreatic cancer [4].

One-carbon metabolism—consisting of folate and methionine cycles—for its optimal function requires substrates and cofactors called methyl donors (such as folate, betaine, choline, methionine and B vitamins), which are bioactive food components and provide methyl groups [5,6]. The abovementioned metabolic cycles are essential in several physiological processes including amino and nucleic acid metabolism, redox defense, cell growth, apoptosis, and adequate DNA methylation [5,7,8]. Additionally, the imbalanced nutritional status of vitamin B2, B6 and B12 can alter the function of folate and methionine cycles and consequently can cause pathological changes [9]. Accordingly, several studies have discussed the importance of dietary methyl donors in the prevention of and reduction in the risk of cancer development such as breast, colorectal, lung, and pancreatic cancer [10,11,12,13,14,15,16,17,18,19]. Additionally, both clinical trials and in vivo animal studies have demonstrated that methyl-donor micronutrient intake can reduce the risk of development of several cancer types by altering DNA methylation status [7,9]. Moreover, a pilot study reported that a high dose of vitamin B6 may enhance the antitumor activity of standard chemotherapies in the case of colorectal, pancreatic, and esophageal carcinomas [20].

In relation to prevention, the interrelationship between methyl donors is also an important aspect of the metabolism. A case-control study nested within the European Prospective Investigation into Cancer and Nutrition (EPIC) revealed that higher plasma concentrations of choline and betaine may compensate low plasma levels of folate, and consequently may reduce the risk of colorectal cancer [21]. Moreover, an in vivo study found close relation between choline, methionine and folate, implying that all of these three nutrients should be examined together when studying their effects [22].

Nuclear factor kappa-B (NFkB) has a central role in the network of cell signaling as it is present in every cell, controls over 150 genes and several biological processes including cell survival, proliferation, apoptosis, stress response, innate and adaptive immune response, and there is considerable evidence of its involvement in many human diseases, including cancers. NFkB regulates proliferation/cell cycle in the pancreas and is considered to balance between proliferation and apoptosis; therefore, it is a key regulator towards malignant growth [23]. The inhibition of NFkB decreased the expression of several EMT transcription factors, mesenchymal markers and invasion in vitro, as well as tumor volume, EMT gene expression, and overall metastasis, while restoring cell–cell junctions in vivo [24].

After the liberation of NFkB dimer from inhibitory kappa B (IkB) following the inhibitory kappa B kinase (IKK) activation, its nuclear localizing signal appears and allows translocation to the nucleus, where the nuclear factor binds to enhancer elements to gene promoters. The post-translational modification of the NFkB dimer, such as phosphorylation, is another layer of NFkB signaling, and either can enhance or reduce the effect of NFkB [25].

NFkB provides a mechanistic link between inflammation and cancer [26], where IL-17 can activate the canonical NFkB signaling pathway [27], and switches on the pro-tumorigenic program in pancreatic cancers [28]. Chronic inflammation causes elevated constitutive activity of NFkB with the release of cytokines such as TNFa, IL-1b, IL-6 IL-8 and subsequently STAT3 activation [25]. STAT3 and NFkB synergistically control common sets of genes encoding cytokines and chemokines [29]. Additionally, they can recruit each other to their own gene promoters facilitating their own activation [30], i.e., unphosphorylated STAT3 can bind and activate the transcription of NFkB [29].

Moreover, STAT3 can modify NFkB post-translationally by acetylation affecting the prolongation of nuclear retention and the activity of NFkB, which play roles in cancer development [30]. The overexpression of STAT3 itself a prognostic factor in several cancers including pancreatic ductal adenocarcinoma (PDAC) [31]. Additionally, STAT3 inhibition results in delayed G1/S phase with increased p21^WAF1/Cip1^, and the small hairpin RNA for silencing the expression of STAT3 and RNA interference results in inhibited cell proliferation, cell cycle arrest, tumorigenicity, vascularization, motility, invasion, and in increased apoptosis [32].

Previously, we explored the effect of methyl donors on cell cycle and apoptosis in breast and lung cancer cell lines [33]. In our present study, we aimed to investigate these effects on a Panc-1 adenocarcinoma cell line as well. Additionally, as pancreatic cancer development is associated with NFkB-related inflammation and high metastatic potential [26], as well as chemoresistance [34], we also aimed to examine whether methyl donors could affect:Some aspects of inflammatory processes influenced by pancreatic cancer cells;The expression levels of related metastatic markers in Panc-1 cancer cell line.

## 2. Results

### 2.1. Methyl Donors Affect Tumor Cell Proliferation

MTS proliferation assay was used to detect the effect of methyl-donor treatments on the growth of Panc-1 cancer cell line. Both concentrations of methyl donors used significantly reduced the proliferation rate at 48 h (*p* < 0.01 at both concentrations) and 72 h (*p* < 0.001 and *p* < 0.01 at 1× and 20× concentrations, respectively) compared to the non-treated control group (Figure 1A).

### 2.2. The Effect of Methyl Donors on Cell Cycle

Changes in SubG1 fraction, indicating apoptotic cells as well, were measured by flow cytometry after methyl-donor treatments. A significantly increased SubG1 fraction of Panc-1 cells was detected after 20× methyl-donor treatments (*p* < 0.05) compared to control (Figure 1B,C).

Therefore, we investigated the level of p21^WAF1/Cip1^ growth inhibitory protein, which is a mediator of p53-dependent G1 phase arrest, using Western blot analysis. We found significantly increased levels of p21^WAF1/Cip1^ protein (*p* < 0.001) at the 20× methyl-donor concentration after 48 h treatment compared to control (Figure 1D,E).

It is known that the activation of the MAPK/ERK signaling pathway is required to induce genes involved in cell cycle entry and to suppress those genes, which inhibit this process, such as *CDKN1A*, the gene of p21^WAF1/Cip1^ protein. We found significantly decreased p-Erk 1/2 protein levels at 48 h in both the 1× and 20× methyl-donor treated groups (*p* < 0.05 and *p* < 0.001, respectively) measured by Western blot (Figure 1D,E).

### 2.3. Detection of Apoptosis and Related Pathway Elements after Methyl-Donor Treatments

Annexin V positive early apoptotic cell fraction was measured using flow cytometry after methyl-donor treatment, but only a tendency (*p* = 0.326) of increase was seen in Panc-1 tumor cells compared to control. However, a significantly decreased number of live cells was detected after 20× methyl-donor treatments compared to the non-treated control (*p* < 0.01) (Figure 2A,B).

We investigated methyl-donor-induced changes in the expression of pro-and anti-apoptotic proteins using Western blot. The pro-apoptotic Bak, Puma and Caspase-9 protein levels significantly increased at 72 h in the 20× methyl-donor-treated group compared to control (*p* < 0.01, Figure 3A–D). However, we could not detect cleaved Caspase-3. The anti-apoptotic proteins Bcl-2 and Mcl-1 did not change significantly after methyl-donor treatments in Panc-1 cells.

### 2.4. The Effects of Methyl Donors on Metastatic Potential of Panc-1 Cell Line

Pancreatic ductal carcinoma is described as an aggressive tumor with invasive tumor growth and early metastasis; therefore, we checked the protein level of VEGF, SDF-1, MMP-9 by cytokine array (Figure 4A,B) and SDF-1, MMP-9 and E-cadherin by Western blot (Figure 4C,D). We found a significant decrease in the protein level of VEGF and SDF-1a (*p* < 0.05 and *p* < 0.01, respectively) after 48 h treatment with 20× methyl donors compared to the control by cytokine array (Figure 4A,B) and SDF-1a level by Western blot (Figure 4C,D).

Additionally, we found a significant increase in the protein level of E-cadherin after 48 h of treatment with both concentrations of methyl donors compared to the non-treated control group (Figure 4C,D).

### 2.5. The Effect of Methyl Donors on Immune-Related Pathways and Molecules

We detected significantly decreased IL-17a cytokine levels (*p* < 0.01) at 48 h of 20× methyl-donor treatment compared to non-treated Panc-1 cells by cytokine array (Figure 5A,B).

Additionally, we found significantly decreased NFkB protein levels (*p* < 0.05) at 48 h of 20× methyl-donor treatment in Panc-1 cells compared to the controls by Western blot (Figure 5C,D). However, we could not reach significance in the decrease in the levels of CD30, either by cytokine array or by Western blot, nor of STAT3 protein measured by Western blot (Figure 5A,D).

### 2.6. The Level of IL-8 and IL-6 Cytokines in Plasma Samples from Pancreatic Cancer Patients

We collected 35 serum samples from pancreatic cancer patients and measured IL-8 and IL-6 pro-inflammatory cytokine levels in their plasma by ELISA. We found significantly increased IL-8 and IL-6 cytokine levels in the plasma of pancreatic cancer patients compared to the healthy control group (*p* < 0.01 and *p* < 0.0001, respectively) (Figure 6A,B).

## 3. Discussion

Previously, we described that methyl donors are able to induce apoptosis and attenuate proliferation pathways in breast and lung cancer cell lines [33]. In our present study, we aimed to explore if similar changes could be detectable in Panc-1 cancer cells. Pancreatic cancer is considered a lethal disease with high mortality as a consequence of its high metastatic potential, chemoresistance, and poor prognosis [2,3]. Recent studies suggest that this phenotype is closely related to both the constitutive and induced activation of nuclear factor-kB (NFkB), which gives the mechanistic links of the chronic inflammation to cancer development in pancreatic cancer [34]. Therefore, we also aimed to explore whether and how methyl-donor treatment can influence Panc-1-related inflammation processes, as well as its metastatic potential.

There are several approaches to overcome disease burden and progression, including preventive and complementary therapies, such as diet, besides conventional cancer treatments.

Altered methyl-donor metabolism is connected to pathological disturbances such as heart disease, birth defects, aging, and pancreatic toxicity, but it is mostly studied in relation to cancer development [9]. In a meta-analysis, Bo et al. found that dietary folate intake is associated with decreased risk of all-cause mortality and a wide range of chronic diseases, including cancer [17]. Wei and Mao found that high vitamin B6 intake might be associated with a reduced risk of pancreatic cancer [19]. Moreover, another meta-analysis assessing only case-control studies observed an inverse association between the high intake of folate and pancreatic cancer [18]. Methyl donors such as B vitamins are required for energy-yielding metabolism, oxygen transport, and neuronal functions, and these are essential in optimal cognitive and psychological states; therefore, they could support conventional therapy in cancer patients by overcoming mental and physical fatigue [35,36].

We found that methyl-donor treatment significantly decreased the proliferation of Panc-1 cells along with p-Erk 1/2 level, and significantly increased the p21^WAF1/Cip1^ cyclin-dependent kinase inhibitor level as well. Although p21^WAF1/Cip1^ is a growth inhibitory protein, which is a mediator of p53-dependent G1 phase arrest, the *TP53* gene is mutated in Panc-1 cells; thus, we expected a p53-independent upregulation of this protein induced by methyl donors in this cell line [37]. It is also known that the moderate activation of the MAPK/ERK signaling pathway is required to induce genes involved in cell cycle entry and to suppress those genes that inhibit this process, such as p21^WAF1/Cip1^ [38]. The p21^WAF1/Cip1^ expression is inversely associated with tumor differentiation, clinical stages, and lymph node metastasis in pancreatic cancers [39]. Therefore, we suppose that the decreased p-Erk 1/2 expression opened the way to the increased p21^WAF1/Cip1^ expression level, and resulted in a significantly decreased proliferation rate after methyl-donor treatment.

We found significantly increased subG1 fraction in cell cycle analysis, which indicates apoptotic cells; therefore, we investigated the intrinsic apoptotic pathway based on our previous study [33]. We found significantly increased pro-apoptotic Bak, Puma and Caspase-9 protein levels after methyl-donor treatment, but not in the case of the cleaved Caspase-3 level. This, however, could explain why we could not detect significantly increased Annexin-V-positive cells, neither in early nor in late apoptotic events after methyl-donor treatment by flow cytometry, although there is caspase-independent pathway in mitochondria-initiated apoptosis through the activation of AIF as well [40]. It is also known that the apoptosis inhibitor cIAP-2, survivin, livin and XIAP are overexpressed in pancreatic adenocarcinoma and are implicated in resistance to chemotherapy, and NFkB has been described to upregulate these genes [41].

Inflammation contributes to the development of several diseases. Cancers, including pancreatic cancer, are also known as inflammation-related disorders; additionally, chronic pancreatitis has been shown as a predisposition to pancreatic cancer [23,34,42]. NFkB has significant role in the activation of pro-inflammatory immune response, in promoting metastasis and chemoresistance [34], and it is constitutively activated in around 70% of pancreatic cancers [23,24]. IL-17 can activate the canonical NFkB signaling pathway [27] and switches on the pro-tumorigenic program in pancreatic cancers [28]. Parallel with these findings, we also found significantly increased pro-inflammatory cytokine IL-6 and IL-8 from pancreatic cancer patients compared to control, indicating the high probability of an activated NFkB pathway in these patients.

Interestingly, after methyl-donor treatments in our in vitro experiments, we found significantly decreased NFkB as well as IL-17a cytokine levels. This suggests that an appropriate amount of methyl donors in the diet might be an effective complementary and also preventive therapy, from which pancreatic cancer patients with increased NFkB and IL17 level, as well as healthy people, could benefit to keep their inflammation level under control. Some researchers have investigated p-NFkB as well; however, this is a post-translational modification of the NFkB, and another layer of this signaling, which was out of scope of our project.

As it is known that the overexpression of STAT3 is a prognostic marker in several tumors, including PDAC [31], and that NFkB and STAT3 signaling are connected with each other at several levels, affecting cell cycle and apoptosis [29,30], we aimed to only measure the protein level of STAT3, itself a prognostic factor in PDAC, without its activation level (p-STAT3). Although we could not demonstrate significance in the decrease in the STAT3 level after a single dose of methyl-donor treatment in vitro, we have seen this in case of IL-17 and NFkB. As we also found significantly increased levels of IL-8 and IL-6 cytokine in the plasma of pancreatic cancer patients, we suppose this was probably through IL17/NFkB/STAT3 signaling. These findings suggest that PDAC patients could benefit from a well-balanced methyl-donor-containing diet reducing the activity of the above-mentioned signaling path, and thus the related inflammation level, with several physiological benefits.

CD30 is a rediscovered target of new immunotherapies in non-lymphomatous solid tumors [43]. It triggers anti-apoptotic and pro-survival functions [44] through partly activating the NFkB pathway as well as mitogen-activated protein kinase (MAPK) pathways, including ERK1 and ERK2 [45]. As part of our investigation of inflammation processes after methyl-donor treatment, we aimed to measure the changes of CD30. However, we could not reach significance regarding the decrease in CD30 after methyl-donor treatment, neither by cytokine array nor by Western blot.

As pancreatic cancers are described as a malignancy with high metastatic potential, we aimed to detect if methyl donors could change this phenotype. We found the methyl-donor treatment could significantly decrease the SDF-1 and VEGF level, while significantly increasing the expression level of E-cadherin, a cell adhesion molecule, which is a marker of normal polarized epithelial cells as well as of well-differentiated tumor cells [46]. However, we could not detect a significant decrease in MMP-9 after methyl-donor treatment compared to controls. Stromal cell-derived factor 1 (SDF-1) is expressed both in stromal and cancer cells, and has a well-known role in pancreatic tumor cell migration and angiogenesis [47]. Its elevated level is responsible for poor prognosis [48]. Additionally, SDF-1 can protect cancer cells from drug-induced apoptosis acting on NFkB or, indirectly, modulating tumor cell adherence [49].

Some researchers have questioned the relevance of the application of methyl donors even from a dietary source, implying that, in Nicotinamid N-methyl-transferase (NNMT)-overexpressing tumors, methyl donors might fuel the NNMT enzyme causing a worse prognosis. They refer to studies where silenced NNMT has resulted in decreased migration, invasion, proliferation, and increased chemosensitivity [50], and also that NNMT expression is enhanced in cancer stem cells [51]. The NNMT enzyme takes a methyl group from S-adenosyl-L-methionine (SAM) and adds it to nicotinamide (NA), generating methyl-nicotinamid (MNA) and S-adenosyl-L-homocysteine as a result [52]. It is believed that it would decrease the methylation capacity of the methionine cycle towards the DNA/histone methylation path, causing aberrant methylation, which could lead to cancer progression in the case of the overexpression of NNMT. However, an earlier study from 2013 described that NNMT-overexpressed and silenced cells showed enhanced or reduced migration/invasion, respectively, if the cells were grown in methionine-low conditions, while those in methionine-high medium did not show differences in these phenotypes. This group also reported that NNMT, which is overexpressed in a diverse set of cancers, regulates the protein methylation state of tumor cells through a distinct mechanism that involves altering ratios of SAM:SAH [53]. These results also support our hypothesis, and our findings as well, suggesting that balanced nutritional status including methyl donors can revert healthy states of the cells, even with altered, overexpressed protein conditions, such as NNMT overexpression.

In conclusion, we demonstrated that methyl-donor treatments are able to diminish the proliferation of Panc-1 cells, possibly by decreasing the MAPK/ERK pathway and through increased levels of p21^WAF1/Cip1^ growth inhibitory protein, along with significantly increased SubG1 fractions. Moreover, methyl-donor treatments were able to induce the intrinsic apoptotic pathway, and reduce the level of IL-17a with related NFkB. Consequently, this attenuated several unfavorable phenotypes of the Panc-1 cell line, such as loss of cell polarization by significantly increased E-cadherin expression, and high VEGF and SDF-1 expression by their significantly decreased expression. Therefore, we can conclude that adequate amounts of methyl donors could initiate the restoration of normal cell-like phenotype and function in Panc-1 cancer cells, and thus might be a reasonable tool in a dietary intervention or alternative treatment applied in pancreatic cancer patients. However, additional and well-designed studies are needed to confirm its application in clinical settings.

## 4. Materials and Methods

### 4.1. Human Samples

Anticoagulated (K3-EDTA) whole blood samples were taken from 35 pancreatic cancer patients and 9 non-cancerous controls based on the research project approved by the Hungarian Ethical Committee of Scientific Research (No. 28123-6/2019/EÜIG). Plasma samples were collected after centrifugation to detect the levels of IL-6 and IL-8 cytokines by ELISA.

### 4.2. Cell Culture Conditions

Panc-1 adenocarcinoma cell line was purchased from American Type Culture Collection (ATCC, Manassas, VA, USA), and was cultured in Dulbecco’s modified Eagle medium (DMEM with 4.5 g/L glucose, BE12-604Q; Lonza, Basel, Switzerland), and supplemented with 10% fetal bovine serum (FB-1090; Biosera, Nuaille, France) and 0.4% gentamycin (Sandoz, Basel, Switzerland; 80 mg/2 mL). Cells were kept at 37 °C in a humidified atmosphere of 5% CO_2_. Cells were tested for Mycoplasma sp. applying the methodological article written by Uphoff et al. [54]. Only negative cell line was used for research purposes.

### 4.3. Methyl-Donor Treatment

Panc-1 cells were treated with different concentration (1×, 20×) mixtures of methyl donors based on a previous study by Park et al. [55]. Basal concentration (1×) was: 17 mg/L L-methionine, 9 mg/L choline chloride, 3 mg/L folic acid, and 2 mg/L vitamin B12, while the 20× concentration was 20 times more of the 1×. L-methionine, choline chloride, folic acid, and vitamin B12 were purchased from Sigma Aldrich (M5308, C7527, F8758, V6629, respectively; St. Louis, MO, USA).

### 4.4. Cell Proliferation Assay

Cells were plated onto 96-well plates at a cell density of 3 × 103 cells/mL. When cells reached 50% confluence, culture media were changed to methyl-donor-supplemented media and incubated for 24–72 h. Cell growth was measured by a colorimetric MTS (3-(4,5-dimethylthiazol-2-yl)-5- (3-carboxymethoxyphenyl)-2-(4 sulfophenyl)-2H-tetrazolium, inner salt) cell proliferation assay (CellTiter 96^®^AQueous One Solution Cell Proliferation Assay, G3582; Promega, Madison, WI, USA) according to the manufacturer’s instructions. Briefly, 20 µL MTS reagent was added into 100 µL of culture medium in each well at 24 h, 48 h and 72 h of treatments. After 2 h of incubation, the absorbance of the soluble formazan product was measured at 490 and 690 nm with a plate reader (Labsystems Multiskan MS, Thermo Fisher Scientific, Waltham, MA, USA).

### 4.5. Cell Cycle Measurement

Cells were plated onto 6-well plates at a cell density of 3 × 104 cells/mL. After 24, 48, and 72 h treatments of methyl donors, cells were washed with PBS then trypsinized using 1× trypsin-EDTA solution (XC-T1717; Biosera, Nuaille, France). After washing steps, cells were fixed in ice-cold 70% ethanol at room temperature for 20 min, then were kept at −20 °C for an additional 30 min. Cells then were washed twice in PBS. After resuspension in PBS containing 1% RNaseA (R5503; Sigma Aldrich, 10 mg/mL) and 20 µL propidium iodide solution (P3566, Thermo Fischer Scientific Inc, Waltham, MA, USA; 1 mg/mL), samples were incubated for 1h at 4 °C. Cell cycles were detected by the CytoFLEX flow cytometer using CytExpert software (Beckman Coulter, Indianapolis, IN, USA).

### 4.6. Detection of Apoptosis

Panc-1 cells were plated and treated as described above in the section of ‘Cell Cycle Measurement’. After centrifugation (1500 rpm, 5 min), apoptotic cell fraction was determined by FITC Annexin V Apoptosis Detection Kit with PI (640914; BioLegend, San Diego, CA, USA). Annexin V and/or PI positive cell fractions were detected by the CytoFLEX flow cytometer using CytExpert software (Beckman Coulter, Indianapolis, IN, USA).

### 4.7. Western Blot Analysis

Cells were lysed in radioimmunoprecipitation assay (RIPA)-buffer, supplemented with 0.5 mM Na-orthovanadate, 10 mM NaF, and 1:200 Protease Inhibitor Cocktail (P8340, Sigma-Aldrich, St. Louis, MO, USA). Lysates were collected in tubes then centrifuged on 12,000 rpm for 15 min. Total protein concentration was determined using Pierce Rapid Gold BCA Protein Assay Kit (A53226; Thermo Fisher Scientific, Waltham, MA, USA). Cell extracts were mixed with 5× sample loading buffer containing 2-mercaptoethanol (1610710; Bio-Rad, Hercules, CA, USA) and were incubated at 95 °C for 5 min. For Western blot analysis, 30 or 20 µg of total proteins were loaded and run in 10% sodium dodecyl sulphate polyacrylamide gel (SDS-PAGE) at 80 V for 20 min, then at 180 V for 50 min on Mini Protean vertical electrophoresis equipment (Bio-Rad, Hercules, CA, USA). Proteins were transferred onto Immobilon-P PVDF transfer membrane (IPVH00005; Merck KGaA, Darmstadt, Germany) by blotting at 100 V for 60 min at +4 °C. Membranes were blocked, then incubated with primary antibodies overnight at +4 °C (p-Erk1/2 (1:2000, Cat. No. 4370, Cell Signaling Technology—CST), Bak (D4E4) (1:1000, Cat. No. 12105, CST), Puma (1:1000, Cat.No 12450, CST), CD30 (1:1000, Cat. No. 25114; CST), NFkB (1:1000, Cat. No. 8242, CST), Caspase-9 (1:1000, Cat. No. 9502, CST), cleaved Caspase-3 (Asp175) (1:1000, Cat. No. 9661, CST), SDF-1a (FL-93) (1:500, Cat.No sc-28876, Santa Cruz Biotechnology—SCB), STAT3 (1:500, Cat. No. RB-9237-PO, NeoMarkers), p21 (SX118) (1:200, Cat. No. sc-53870, SCB), MMP-9 (GE-213) (1 µg/mL, Cat. No. MAB13415, Sigma-Aldrich) and E-cadherin (0.5 µg/mL, Cat. No. AF648, R&D Systems)). β-actin (13E5) (1:5000, Cat. No. 4970S, CST) was used as loading control. After washing steps, membranes were incubated with horseradish peroxidase (HRP)-labelled secondary antibodies for 60 min at room temperature. Goat anti-rabbit IgG (1:1000, Cat. No. 7074S, CST), horse anti-mouse IgG (1:1000, Cat. No. 7076S, CST) and rabbit anti-goat IgG (1:1000, Cat. No. P0449, Dako) secondary antibodies were applied. Immunodetection was performed using SuperSignal West Pico Chemiluminescent Substrate Kit (34080; Thermo Fisher Scientific,Waltham, MA, USA). Membranes were visualized by iBright FL1500 Imaging System (Thermo Fisher Scientific, Waltham, MA, USA). Densitometric analysis of the immunoblots was performed using Image J software (developed by National Institute of Health (NIH) and Laboratory for Optical and Computational Instrumentation (LOCI), University of Wisconsin).

### 4.8. Detection of Inflammatory Cytokine Levels in Panc-1 Cell Line

Panc-1 cell supernatants were collected after methyl-donor treatment and stored at −20 °C until analysis. The relative expression of inflammatory cytokines was determined using Proteome Profiler Human XL Cytokine Array Kit (ARY022B, R&D Systems, Minneapolis, MN, USA) according to the manufacturer’s instructions. Membranes were visualized by iBright FL1500 Imaging System (Thermo Fisher Scientific, Waltham, MA, USA). Analysis of mean spot pixel density was executed using Image J software (developed by National Institute of Health (NIH) and Laboratory for Optical and Computational Instrumentation (LOCI), University of Wisconsin).

### 4.9. Quantification of IL-6 and IL-8 Cytokine Levels in Pancreatic Cancer Patients

Blood samples from patients with pancreatic cancer were collected in blood collection tubes containing K3-ethylenediaminetetraacetic acid (K3-EDTA). Plasma was separated by centrifugation at 3000 rpm for 10 min at 4 °C. Aliquots of each plasma sample were stored at −20 °C until analysis. Enzyme-linked immunosorbent assay (ELISA) was performed to quantify plasma concentrations of IL-6 and IL-8 using the Quantikine Human ELISA Kit (D6050, D8000C, respectively; R&D Systems, Minneapolis, MN, USA) according to the manufacturer’s instructions. Optical density of each well was measured at 450 and 570 nm with a microplate reader (Labsystems Multiskan MS, Thermo Fisher Scientific Waltham, MA, USA). Plasma concentrations were calculated from the standard curves and were presented in pg/mL.

### 4.10. Statistical Analysis

All experiments were repeated at least *n* = 3 different times and are expressed as mean ± standard deviation (mean ± SD), or median ± interquartile range. We applied one-way and two-way ANOVA with Bonferroni post-tests for multiple comparisons applying GraphPad Prism software (GraphPad Software LLC, San Diego, CA, USA). The significance levels were used as *: 0.01 < *p* < 0.05, **: 0.001 < *p* < 0.01, and ***: *p* < 0.001 in all experiments except for ELISA, where the *p* value style of GP Prism (*: 0.01 < *p* < 0.05, **: 0.001 < *p* < 0.01, *** 0.0001 < *p* < 0.001 and ****: *p* < 0.0001) were applied.

## Figures and Tables

**Figure 1 ijms-23-02546-f001:**
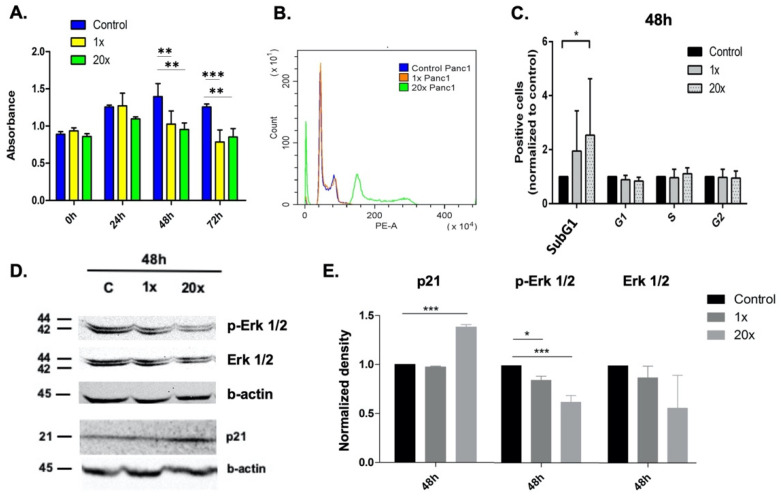
Proliferation and cell cycle analysis of Panc-1 cells after methyl-donor treatment. Proliferation of Panc-1 cells decreased significantly both at 48 h and 72 h after 1× and 20× methyl-donor treatments (**A**). SubG1 fraction of Panc-1 cells were significantly increased after 20× methyl-donor treatments at 48 h compared to untreated control (**B**,**C**). p21 (p21^WAF1/Cip1^) growth inhibitory protein significantly increased, while p-Erk 1/2 protein level significantly decreased after both 1× and 20× methyl-donor treatments at 48 h compared to untreated control (**D**,**E**). Each bar represents the average number of positive cells or normalized density in cell cycle analysis or Western blot, respectively; from at least 3 repeats ± SD. Statistical significance: *: *p* < 0.05; **: 0.01< *p* < 0.05; ***: *p* < 0.001. 1× and 20×: concentrations of methyl donors.

**Figure 2 ijms-23-02546-f002:**
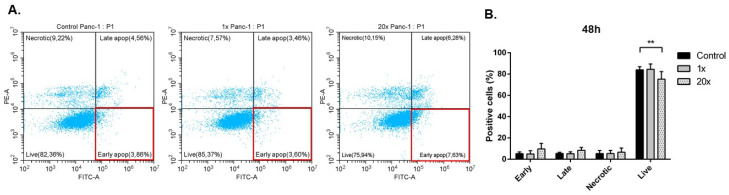
Detection of apoptosis in Panc-1 cell line. Early apoptotic cells (red-squared highlighted areas) did not change significantly; however, the number of live cells (lower left square) decreased significantly (**A**,**B**). Each bar represents the average percentage of positive cells in early apoptotic, late apoptotic, necrotic, and live cells area from at least 3 repeats ± SD. Statistical significance was plotted as **: *p* < 0.01. Early: early apoptotic cells; Late: late apoptotic cells; Necrotic: necrotic cells; Live: live cells. 1× and 20×: concentrations of methyl donors.

**Figure 3 ijms-23-02546-f003:**
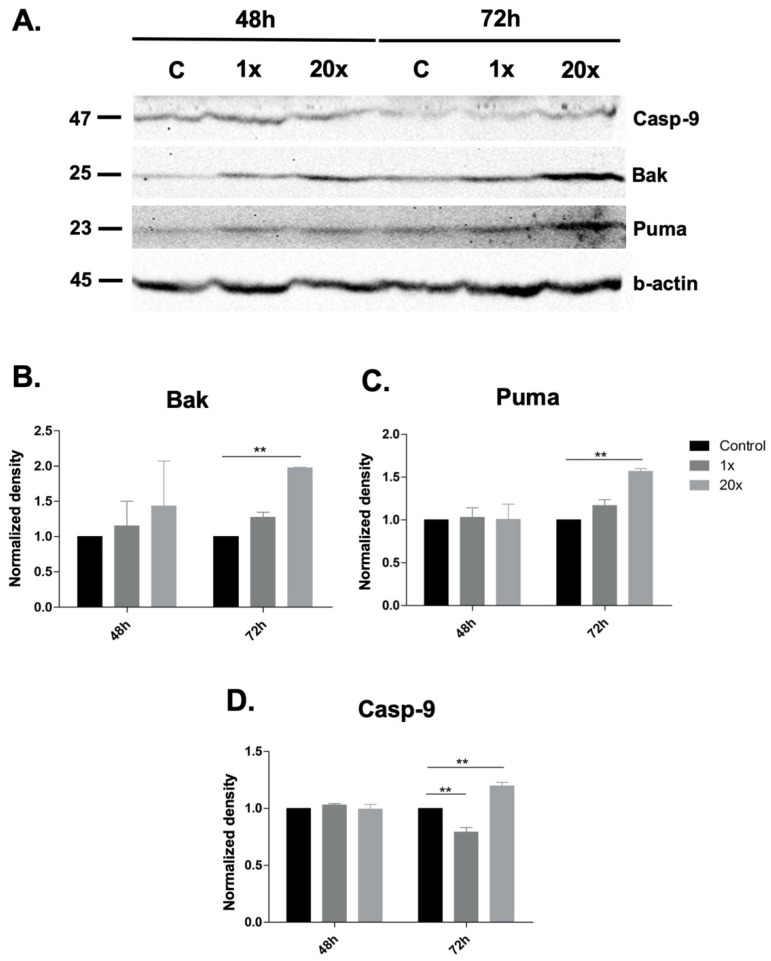
Western blot analysis of apoptotic proteins in methyl-donor-treated Panc-1 cells. Pro-apoptotic Bak, Puma and Caspase-9 were significantly increased in Panc-1 cells after 72 h of 20× methyl-donor treatment (**A**–**D**). Each bar represents the average normalized density from at least 3 repeats ± SD. Statistical significances plotted as **: *p* < 0.01. 1× and 20×: concentration of methyl donors.

**Figure 4 ijms-23-02546-f004:**
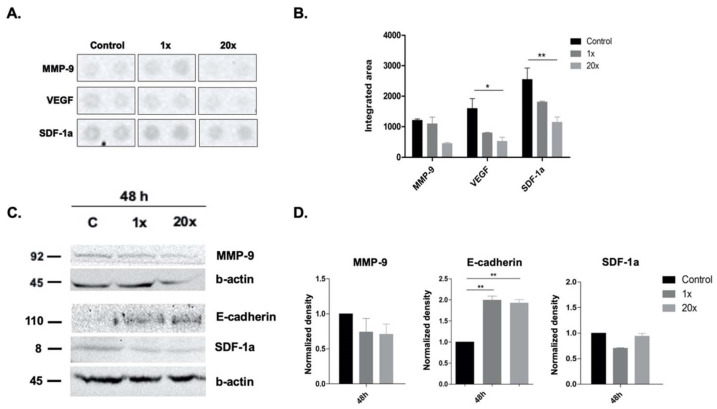
Cytokine array and Western blot analysis of metastatic potential-related markers in Panc-1 cell line. Significantly decreased VEGF and SDF-1a level by cytokine array and SDF-1a level by Western blot were detected in Panc-1 cells after 48 h of 20× methyl-donor treatment (**A**–**D**). Expectedly, E-cadherin level changed inversely, and its significant increase was detected after 48 h of 20× methyl-donor treatment (**C**,**D**). Unexpectedly, we could not measure significantly decreased MMP-9 level after the methyl-donor treatments. Each bar represents the average normalized density from at least 3 repeats ± SD. Statistical significance are plotted as *: *p* < 0.05; **: *p* < 0.01. 1× and 20×: concentrations of methyl donors.

**Figure 5 ijms-23-02546-f005:**
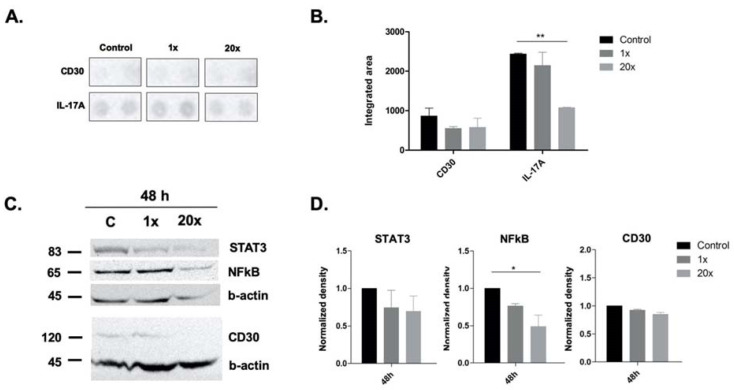
Cytokine array and Western blot analysis of inflammation-related markers in Panc-1 cell line. Significantly decreased level of IL-17a measured by cytokine array and NFkB by Western blot were detected after 48 h of 20× methyl-donor treatment (**A**–**D**). Decreased protein level of CD30 did not reach significance after 48 h methyl-donor treatment, either by cytokine array (**A**,**B**) or by Western blot, similar to STAT3 protein level (**C**,**D**). Each bar represents the average normalized density from at least 3 repeats ± SD. Statistical significance are plotted as *: *p* < 0.05; **: *p* < 0.01. 1× and 20×: concentration of methyl donors.

**Figure 6 ijms-23-02546-f006:**
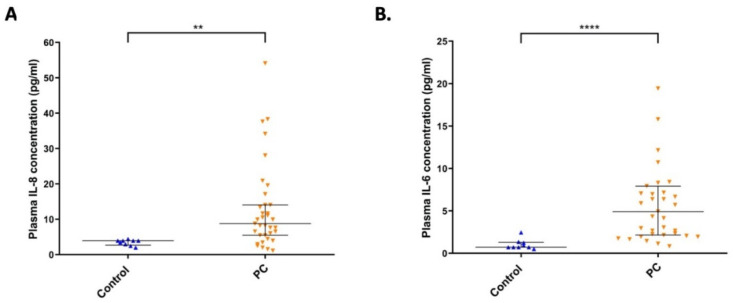
Measurement of plasma IL-8 and IL-6 cytokine levels of pancreatic cancer patients. Significantly increased levels of IL-8 and IL-6 were detected by ELISA from 35 pancreatic cancer patients compared to non-cancerous controls (**A**,**B**). Each dot represents a concentration of either IL8 or IL-6 cytokine level in a pancreatic cancer patient where boxes show the median value with the interquartile range. Statistical significances are plotted as **: *p* < 0.01; ****: *p* < 0.0001.

## Data Availability

Not applicable.

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
