# Peer review of "Methyl Donors Reduce Cell Proliferation by Diminishing Erk-Signaling and NFkB Levels, While Increasing E-Cadherin Expression in Panc-1 Cell Line"

_ijms, 2022, doi:10.3390/ijms23052546_

Round 1
Reviewer 1 Report
In this study the authors investigated the roles of methyl-donors in improving pancreatic cancer treatment. They found that methyl-donor treatment significantly increased p21WAF1/Cip1 cyclin dependent kinase inhibitor, pErk1/2 levels and the proliferation rate. Methyl donor treatments also increased the pro-apoptotic protein Bak, Puma and Caspase-9, and reduced IL-17a and NFkB levels. A significant decrease of VEGF and SDF-1a levels and increased E-cadherin expression was also detected after methyl-donor treatment.
Although the manuscript is well written, it needs substantial improvement:
- section 2.1. Methyl-donors affect tumor cell proliferation: a rappresentative image of MTS must be reported
- Figure 1C-D: How did the authors normalyzed pERK expression? Did they used ß-Actin? phosphorylated proteins should be normalized with the unforphorylated form (in this case total ERK1/2) because changes found in the phosphorylated form could be due to increased/decreased expression of the total form rather than a change in phosphorylation and then activation.
- Figure 3: a representative blot of Cleaved Caspase 3 must be reported. That one showed in figure 3A does not represent the results found by the authors
- Lines 226-238: although authors discussed activation of NFkB and STAT3 signaling they only evaluated the total STAT3 and NFkB. To evaluate a role of methyl donors in modulating STAT3 and NFkB signaling the authors should also investigate their phosphorylated forms.
- In the discussion section, authors underline that methyl-donor treatments are able to reduce the aggressiveness of cancer cells and they suggest that administering methyl-donors could be a dietary intervention or alternative treatment for cancer patients. However, it is well known that several methyltransferases, as Nicotinamide N-methyltransferase, are actively involved in solid tumors progression (PMID: 34018676; PMID: 27901497) or even crucially involved in cancer stem cells biology (PMID: 32150326) and there is an active research focused on blocking their activity (PMID: 34424711). Thus, on one hand, administering methyl-donors may look like beneficial, but on the other hand, the risk is to fuel these enzymes whose activity is harmful for the patient outcome. Please widely discuss this crucial point in the discussion section.
Author Response
Author's Reply to the Review Report (Reviewer 1)
Thank you for your comments and suggestions for correction of our manuscript. Please see our detailed answers and changes (in red) in the revised manuscript as listed below.
- section 2.1. Methyl-donors affect tumor cell proliferation: a representative image of MTS must be reported
We inserted a representative image for the proliferation into the Figure 1 as part A. and changed the related texts as well (line 78, 83,85-87, 88-90, 97 and 101-102).
- Figure 1C-D: How did the authors normalyzed pERK expression? Did they used ß-Actin? phosphorylated proteins should be normalized with the unforphorylated form (in this case total ERK1/2) because changes found in the phosphorylated form could be due to increased/decreased expression of the total form rather than a change in phosphorylation and then activation.
We did the normalization to the b-actin as it was the loading control. However, we rerun the western blots and redeveloped the p-Erk and newly developed the total Erk as well. We normalized both to the loading control b-actin and made the statistical analysis comparing the treated groups with the untreated control. We found significant changes now adding the data to the previous ones at both concentrations of methyl-donor treatments, while the total Erk did not changed significantly at neither concentration. We added the new lines and stats to the Figure 1 D-E and added texts to describe the new statistical significance as well.
- Figure 3: a representative blot of Cleaved Caspase 3 must be reported. That one showed in figure 3A does not represent the results found by the authors
We rerun the blots and developed them for cleaved Caspase-3 as well, however we did not detect any signal (similarly, as we could any not in our previous study). We have to assume that the presented cleaved Caspase-3 was mislabelled and so placed in the paper mistakenly. We corrected both in the Figure 3.: deleted the lines of the cleaved Casp-3 in part E, and deleted the texts from the figure legend as well, additionally we corrected our findings in both the result and discussion sections (line 119, 218).
- Lines 226-238: although authors discussed activation of NFkB and STAT3 signaling they only evaluated the total STAT3 and NFkB. To evaluate a role of methyl donors in modulating STAT3 and NFkB signaling the authors should also investigate their phosphorylated forms.
We rerun the blots but could not detect p-NFkB. However, we added a better picture for NFkB to the Figure 5. The significance did not change after adding this run to the previous data when preform the statistical analysis.
Both the overexpression and activation (phosphorylation) of STAT3 is known to responsible for several processes, which promote tumour progression (i.e. https://doi.org/10.1186/s13046-019-1206-z), therefore both the detection of decreased expression of STAT3 and decreased phosphorylation level have impact on a possible effective cancer treatment.
We added a better picture to the Figure 5 for STAT3 as well, the significance did not change, however, similarly to NFkB.
- In the discussion section, authors underline that methyl-donor treatments are able to reduce the aggressiveness of cancer cells and they suggest that administering methyl-donors could be a dietary intervention or alternative treatment for cancer patients. However, it is well known that several methyltransferases, as Nicotinamide N-methyltransferase, are actively involved in solid tumors progression (PMID: 34018676; PMID: 27901497) or even crucially involved in cancer stem cells biology (PMID: 32150326) and there is an active research focused on blocking their activity (PMID: 34424711). Thus, on one hand, administering methyl-donors may look like beneficial, but on the other hand, the risk is to fuel these enzymes whose activity is harmful for the patient outcome. Please widely discuss this crucial point in the discussion section.
Nicotinamide N-Methyltransferase are supporting the metabolic switch in a cancer cell (https://www.mdpi.com/2218-273X/11/10/1418), and play role in the glucose metabolism. As it is described in the cited paper above in the 2.3 section “Increased amounts of NNMT alter the methylation status of the genome by reducing available SAM,”, which means this disrupts the one-carbon metabolism, which cycle is working for appropriate methylation of DNA.
Although NNMT and methyl-donors are both transfer methyl group, but in different pathways and working completely in an opposite way.
Methyl-donors keep the one-carbon cycle to work, NNMT disrupts the function of one-carbon cycle.

Reviewer 2 Report
In this paper, the authors studied the effects of methyl-donors on cells and claim that the cell cycle was arrested in Panc-1 cells by methyl-donors. However, I believe that the cell cycle is not really affected, and at most only about 10% of the cells died. Unfortunately, the molecular mechanism is also unknown. I don't know why the authors used this concentration of methyl donor. This paper is not only inadequate in its analyses of the results, but also experimentally problematic. Specific comments are as follows.
Major points.
- As mentioned above, the authors claimed that the addition of methyl-donors arrested the cell cycle of Panc-1 cells, but the results showed that it did not at all (Figure 1B). At most, only about 10% of the cells died (Figure 2). The title of the manuscript does not support the results. This point should be explained.
- Also, why they use a 20x concentrate is beyond me, even after reading the introduction. What are the physiological implications of a 20x concentrate? What is the situation in humans to reach this concentration? What is the effect of a 10x concentrate is applied to the cells, or even a 50x concentrate? There are so many questions.
- It is obvious that the use of a concentrated solution has some effect on the growth of cells. What about the effect on normal pancreatic cells? The authors have previously studied the effects of methyl donors on other cancer cells, but I wonder if he current report shows any specific effects on pancreatic cancer cells.
- Line 73-78: Why don't the authors provide the detailed data on the effect on cell proliferation? It should be graphed, not just mentioned. I have no idea how much cell proliferation has been reduced.
- Overall, the quality of western blotting is poor. The bands are all connected and not well separated, some are barely visible, and the experiment is clearly poorly done. It is impossible to interpret the results based on this.
- Figure 4A and 5A: Imaging results using the cytokine array kit are almost invisible. The authors should re-perform the experiment.
- Figure 3: Why did the authors detect the full length of caspase 9 instead of the cleaved form? Why is cleaved caspase 3 expressed a lot even in control samples?
- Even the 1x solution increased E-cadherin expression (Figure 4) and decreased cell proliferation (Lines 73-78). However, little effect of 1x solution on p21 or Erk was observed. So, how can we interpret the effect of the amount of methyl donor and the results? The molecular mechanism and the purpose of this paper are also unclear.
- Figure 6: What does the increased inflammation in pancreatic cancer patients have to do with the theme of this paper? How does it relate to the effects of methyl donors? Clearly, this is irrelevant data and should be deleted.
Minor points.
Line 283-288: The authors should also describe the composition of the 20x concentrate.
Author Response
Author's Reply to the Review Report (Reviewer 2)
Thank you for your comment and suggestion for correction our manuscript. Please see our detailed answers and changes in the revised manuscript listed below.
Major points.
- As mentioned above, the authors claimed that the addition of methyl-donors arrested the cell cycle of Panc-1 cells, but the results showed that it did not at all (Figure 1B). At most, only about 10% of the cells died (Figure 2). The title of the manuscript does not support the results. This point should be explained.
We changed the phrase “cell cycle arrest” in the title of the manuscript as it would relate to a decreased G1 phase as well. We only detected increased subG1, which includes apoptotic cells, which process was one of our targets to check in this manuscript. Using “cell cycle arrest”, therefore misleading, and inappropriate use of this expression in our case. We change the title as “Methyl-donors decrease cell proliferation by diminishing Erk-signaling, and NFkB level, while increasing E-cadherin expression in Panc-1 cell line”.
- Also, why they use a 20x concentrate is beyond me, even after reading the introduction. What are the physiological implications of a 20x concentrate? What is the situation in humans to reach this concentration? What is the effect of a 10x concentrate is applied to the cells, or even a 50x concentrate? There are so many questions.
We used a mixture as described by Park (2008 doi: 10.1007/s11626-008-9096-y. PMID: 18498022).
It is suggested to assess choline, methionine and folate together when these are studied, as they tight interrelationship makes highly sensitive the folate-mediated one-carbon metabolism (FOCM) to nutrition status, and thus can alter the network output (Niculescu 2002 doi: 10.1093/jn/132.8.2333S. PMID: 12163687, Scotti 2013 DOI: 10.1002/wsbm.1209 PMID: 23408533).
Also, majority of the clinical trials have shown that of methyl-donor micronutrient intake reducing the risk of several cancer types by affecting DNA methylation.
Additionally, a meta-analysis revealed that folate is associated with decreased risk of all-cause mortality and a wide range of chronic disease (Bo 2020 0.3389/fpubh.2020.550753 PMC7770110), and another review with a result, where a low or deficient folate status was associated with several cancers, except prostate cancer, which prevalence was linked with folic acid (FA) supplementation and higher serum level (Pieroth 2018 10.1007/s13668-018-0237-y PMC6132377). Although there are unclear aspects between the in vitro and in vivo studies, majority of clinical trials have shown that methyl-donor micronutrient intake reducing the risk of several cancer types (Mahmoud 2019 10.3389/fonc.2019.00489 PMC6579900). We aimed to explore some possible background processes by applying in in vitro experiments.
There are no data how much of the RDA recommendation (see below) are utilized in humans after taking the daily intake.
This is depending form several factors like intestinal health, other drugs, food with these are taken or taken together, metabolic level, body mass, physical activity and so on. Dietary micronutrients are not drugs; therefore, we only, or mostly have empirical experiences of the useful dosage per day, from known deficiency caused diseases.
Applied amounts of methyl-donors per reaction (well) in 1x treatment in or studies:
L-methionine: 51 ug (RDA is 19mg/kg/day for an adult; https://globalrph.com/rda-and-ear-recommendations-for-essential-amino-acids/)
Choline chloride: 27 ug (RDA is 425-550mg/day; https://ods.od.nih.gov/factsheets/Choline-HealthProfessional/)
B9 (folate): 9 ug (RDA 400ug/day for adults)
B12: 6 ug (RDA 2.4ug/day for adults).
- It is obvious that the use of a concentrated solution has some effect on the growth of cells. What about the effect on normal pancreatic cells? The authors have previously studied the effects of methyl donors on other cancer cells, but I wonder if he current report shows any specific effects on pancreatic cancer cells.
We used a mixture as described by Park (2008 see above). Park et al found no effect of methyl-donor mixture on normal cell (MCF10A), however I do not have information about any normal pancreatic cell experiment with methyl-donor treatment and we did not have access to normal pancreatic cells either.
We showed in our recent study that methyl-donors are significantly decreased the NFkB level, and increased E-cadherin. We also shown that these would probably decrease VEGF and IL17 level as well, however we only used cytokine array, a tool for screening and we did not have sources to confirm these changes by Western-blots.
We did not investigate these proteins in our other study, mainly as pancreatic cancers are a different entity in oncology and more connected with inflammation with related therapeutic resistance, therefore this is rather new findings compare to our previous study with lung and breast cancer cell lines.
We did not investigate p53 involvement here as it is mutated in Panc-1 cell, so this is another point, where the two study radically differ from each other.
Not mention the patient results, which are a proof of a pro-inflammatory background in these population, where our result, where the methyl-donor treatments able to reduce this pro-tumorigenic program by altering the influence of the cancer cells on it, could be a great advantage in a clinical setting.
- Line 73-78: Why don't the authors provide the detailed data on the effect on cell proliferation? It should be graphed, not just mentioned. I have no idea how much cell proliferation has been reduced.
We inserted a representative image for the proliferation into the Figure 1 as part A. and changed the related texts as well (line 78, 83,85-87, 88-90, 97 and 101-102).
- Overall, the quality of western blotting is poor. The bands are all connected and not well separated, some are barely visible, and the experiment is clearly poorly done. It is impossible to interpret the results based on this.
We rerun the Western-blots and changed the lines, where we got better pictures: i.e Fig1., Fig 4 and Fig5. We changed the related texts where it was needed.
- Figure 4A and 5A: Imaging results using the cytokine array kit are almost invisible. The authors should re-perform the experiment.
We reperform the experiment and got similar results.
However, we do not expect strong effect from a “nutritional component”, which are naturally part of the human metabolism, metabolic cycles, moreover, after only a single dose treatment.
We think this cytokine array is a good tool for screening, which anyway should be confirmed either Western-blot or other methods as well, what we performed in some cases. Otherwise, we aimed to take up new directions, which would be meaningful to investigate.
- Figure 3: Why did the authors detect the full length of caspase 9 instead of the cleaved form? Why is cleaved caspase 3 expressed a lot even in control samples?
Caspase-9 is an initiator caspase and part of the apoptosome complex, which can amplify the apoptotic signal by triggering Caspas-3 cleavage: “The release of cytochrome c into the cytosol triggers caspase-3 activation through formation of the cytochrome c/Apaf-1/caspase-9-containing apoptosome complex” (Fulda S, Oncogene (2006) 25, 4798–4811; doi:10.1038/sj.onc.1209608)
We rerun the blots and developed them for cleaved Caspase-3 as well, however we did not detect any signal (similarly, as we could not in our previous study). We have to assume the presented cleaved Caspase-3 was mislabelled and so place in the paper mistakenly. We corrected both in the Figure 3. deleted the lines of the cleaved Casp-3 in the figure part E, and from the figure legend as well, additionally we corrected our findings in the text and discussion sections as well (line 119, 218).
- Even the 1x solution increased E-cadherin expression (Figure 4) and decreased cell proliferation (Lines 73-78). However, little effect of 1x solution on p21 or Erk was observed. So, how can we interpret the effect of the amount of methyl donor and the results? The molecular mechanism and the purpose of this paper are also unclear.
We used the proliferation experiments to screen, if there any effect and is yes, then which concentration of the methyl-donors are able to decrease cell proliferation. Then we checked what could be the reason of this decrease. It either could be by apoptotic/necrotic process or cell cycle changes. Therefore, we checked these two scenarios uncover the molecular reason of the effect the treatments.
We rerun the blots and the decrease of the p-Erk level became significant now in both concentrations. We changed the text and Figure 1 adding these results to the analysis as well (line 89, 100-102).
The reason why the p21 not increasing at 1x concentration could be explained by the sentence in line 207-209, which inserted below (more detail could be found in the paper where we cited): “That is also known the moderate activation of the MAPK/ERK signalling pathway is required to induce genes involved in cell cycle entry and to suppress those genes, which inhibit this process like p21WAF1/Cip1 (28).”
- Figure 6: What does the increased inflammation in pancreatic cancer patients have to do with the theme of this paper? How does it relate to the effects of methyl donors? Clearly, this is irrelevant data and should be deleted.
The relevance of samples from pancreatic patient is that this supports our result by proving that there is a high pro-inflammatory background in pancreatic cancer patients, where our findings, the methyl-donor treatment diminishing the pro-tumorigenic inflammatory processes, could be an advantage in a clinical setting.
Minor points.
Line 283-288: The authors should also describe the composition of the 20x concentrate
In line 287-288 we described the 1x concentrate. We cannot see the necessity describing the composition of the 20x one, as it is just simply 20 times more than the 1x.

Round 2
Reviewer 1 Report
- Author response point 4: As shown in other studies (PMID: 29525378, 34771469), Panc1 cells express both pNFkB and pSTAT3. Which antibodies did the authors used? They may not be appropriated for the detection of the phosphorylated forms. Moreover, the total protein amount loaded may be too low
- Author response point 5: Authors completely missed the point. In the discussion section, authors underline that methyl-donor treatments are able to reduce the aggressiveness of cancer cells and they suggest that administering methyl-donors could be a dietary intervention or alternative treatment for cancer patients. Nicotinamide N-methyltransferase utilizes the methyl donor S-adenosyl methionine as methyl donor, methylating the nicotinamide. This pathway has been found to be involved in solid tumors progression (PMID: 34018676; PMID: 27901497) or even crucially involved in cancer stem cells biology (PMID: 32150326), this is the reason why there is an active research focused on blocking their activity (PMID: 34424711). Thus, administering methyl-donors may be beneficial according to what reported by authors in the manuscript, but may also enhance the activity of other harmful enzymes since administering the substrate enhances the activity of an enzyme. Thus, on one hand, administering methyl-donors may look like beneficial, but on the other hand, the risk is to fuel other harmful enzymes. The authors must account this problem in the discussion section since this is a crucial point and eventually they should discuss possible solutions to avoid this problem.
Author Response
Author’s reply to reviewer 1
Thank you for another thoughts, comments and request to improve our manuscript. We coloured red the new parts of the text in our revised manuscript.
For the comments:
- Author response point 4: As shown in other studies (PMID: 29525378, 34771469), Panc1 cells express both pNFkB and pSTAT3. Which antibodies did the authors used? They may not be appropriated for the detection of the phosphorylated forms. Moreover, the total protein amount loaded may be too low.
We did not aim to measure post-translational modification of NFkB, like phosphorylation, although it would give another layer of the NFkB signaling in this regard as well. However, this was out of scope of our study. Similarly, we only would like to check STAT3 protein level, as itself a prognostic factor in PDAC.
We added explanation of this selection in the discussion section in line 285-286 and 289-290.
However, as per request was in the first revision we reblot the membranes with p-NFkB p65 (Cell signaling, Cat No. 3033T, dilution 1:1000), but we could not detect any signal.
For the question there might was a low protein cc we can answer that we used 20ug/sample to run. We decreased the amount of the protein after the revision requested a nicer Western-blots, as we proposed might the higher amount of 30ug caused a wavy line of the protein bands. Indeed, with 20ug we had clean nice bands, so I think this amount is enough and also sufficient enough to detect phosphorylated areas, as we could detected nice p-Erk signal as well.
The reason why we could not detect p-NFkB, we guess the methyl-donor treatments not affecting the phosphorylation state of the NFkB, at least not in these concentrations.
- “Author response point 5: Authors completely missed the point.” etc
I have read all the selected publications, although I do not understand the relevance of the one with PMID: 27901497 as it is a curcumin related manuscript, might was just a mistype in the number.
We inserted a detailed part for NNMT-methyl-donor relation in the discussion section from line 314-331. We think this part give enough references and detail to demonstrate that our hypothesis ” the balanced nutritional status, included methyl-donors, can revert pre-cancerous states of metabolism even with altered, overexpressed protein conditions, like NNMT overexpression” is supported even those finding stated the referenced manuscripts.

Reviewer 2 Report
None.
Author Response
Dear Reviewer 2,
We revised the manuscript and coloured red the newly changed parts of the text.
We think these changes could clear up the previously not well-detailed sections and missing points of the explanations.
Round 3
Reviewer 1 Report
The manuscript can be accepted in the present form